# miR-383-5p Regulates Preadipocyte Proliferation and Differentiation by Targeting *RAD51AP1*

**DOI:** 10.3390/ijms241814025

**Published:** 2023-09-13

**Authors:** Meigui Wang, Jiahao Shao, Xiaoxiao Zhang, Zheliang Liu, Tao Tang, Guanhe Chen, Siqi Xia, Kaisen Zhao, Zhe Kang, Wenqiang Sun, Xianbo Jia, Jie Wang, Songjia Lai

**Affiliations:** 1College of Animal Science and Technology, Sichuan Agricultural University, Chengdu 611130, China; 2Farm Animal Genetic Resources Exploration and Innovation Key Laboratory of Sichuan Province, Sichuan Agricultural University, Chengdu 611130, China

**Keywords:** rabbits, preadipocytes, miR-383-5p, *RAD51AP1*, differentiation, proliferation

## Abstract

Obesity has become a major health problem worldwide, and increasing evidence supports the importance of microRNAs (miRNAs) in its pathogenesis. Recently, we found that miR-383-5p_1 is highly expressed in the perirenal fat of high-fat-fed rabbits, but it is not yet known whether miR-383-5p is involved in lipid metabolism. Here, we used transcriptome sequencing technology to screen 1642 known differentially expressed genes between miR-383-5p mimic groups and miR-383-5p negative control groups. Gene Ontology Resource (GO) and Kyoto Encyclopedia of Genes and Genomes (KEGG) were enriched in the pathway related to lipid metabolism, and glycine biosynthesis, the NOD receptor signal pathway and nonalcoholic fatty liver were significantly enriched. Afterwards, our research results indicated that miR-383-5p can promote the proliferation and differentiation of rabbit preadipocytes, and there is a direct targeting relationship with *RAD51AP1*. Mechanistically, miR-383-5p directly interacts with the lipid metabolism and participates in adipogenesis and lipid accumulation by targeting *RAD51AP1*. In conclusion, our data highlight a physiological role for miRNA in lipid metabolism and suggest the miR-383-5p/RAD51AP1 axis may represent a potential mechanism for controlling lipid accumulation in obesity.

## 1. Introduction

Obesity is now considered to be an epidemic, spreading rapidly not only in developed countries but also in developing countries [1,2]. Obesity has a positive and significant impact on the prevalence of all chronic diseases, especially hypertension, hyperlipidemia, and diabetes [3,4]. In general, obesity is a chronic disease that is difficult to control, often interrupted by treatment failure, and has become a huge public health problem [5]. Obesity is mainly caused by excessive accumulation of lipids in adipose tissue, including the increase in the volume and number of adipocytes [6,7]. Observational studies on humans have shown that patients with high visceral fat have a higher risk of serious complications [8]. For example, excessive perirenal fat will increase the risk of hypertension and coronary heart disease [9]. miRNAs are small noncoding RNAs that typically inhibit the translation and stability of mRNAs, controlling genes involved in cellular processes such as inflammation, cell-cycle regulation, stress response, differentiation, apoptosis, and migration [10,11]. At the molecular level, Zhong et al. [12] found that Genipin increased the expression levels of miR-142a-5p, which bound to 3′untranslated region of Srebp-1c, an important regulator of lipogenesis, which thus led to the inhibition of lipogenesis. miR-125a-5p should be considered as a regulator of glycolipid metabolism in T2DM, which can inhibit hepatic lipogenesis and gluconeogenesis and elevate glycogen synthesis by targeting *STAT3* [13]. Ni et al. [14] revealed the significant role of miR-802/AMPK axis in hepatic lipid metabolism and identified the therapeutic potential of *Sjp40* in treating obesity-related fatty liver. A growing body of recent research indicates that miRNAs are important in the pathogenesis of abdominal obesity, hypertension, coronary heart disease, atherosclerosis, and diabetes [15].

A large number of studies have shown that miR-383-5p can directly target the 3′-untranslated region of some pro-tumor genes to weaken cancer-related processes, including cell proliferation, cell development and the glycolysis pathway [16,17,18]. miR-383-5p is used as a tumor inhibitor in many types of cancer, including breast cancer, gastric cancer, and liver cancer [19,20,21]. Nowadays, miR-383-5p has become a promising therapeutic agent for treating related cancers. In terms of lipid metabolism, global transcriptomics analysis of Huh7 human hepatoma cells overexpressing miR-383-5p revealed enrichment of lipid and cholesterol metabolic processes; miR-383-5p has potential therapeutic effects in regulating liver lipid metabolism [22]. The above studies suggested that miR-383-5p may be involved in lipid metabolism processes, but the specific mechanism remains unclear. Rabbits have a similar lipid metabolism pathway to humans, making them an ideal model for studying human obesity [23]. Therefore, we investigated the role of miR-383-5p in rabbit preadipocyte proliferation and differentiation by overexpressing or inhibiting miR-383-5p.

## 2. Results

### 2.1. miR-383-5p_1 Is Associated with Adipose Development

In our previous research, we found that rabbits fed with a high-fat diet had a higher body weight than those fed with a standard normal diet. In addition, the total weight of perirenal adipose tissue in rabbits fed with a high-fat diet was also higher [24]. So, we selected three heavier 35-day-old rabbits (heavy rabbits) from the high-fat feeding group and three lighter 35-day-old rabbits (light rabbits) from the standard feeding group to verify whether miR-383-5p_1 is related to adipose development. Triglyceride (TG) and cholesterol (TC) are markers of lipid metabolism and were significantly enhanced in the heavy rabbits (Figure 1A,B). After the qRT-PCR experiment, it was found that the expression of miR-383-5p_1 in heavy rabbits was significantly higher than that in light rabbits (Figure 1C). Afterwards, we also validated the relative expression of miR-383-5p_1 in other tissues. The results showed that miR-383-5p_1 had a higher expression in adipose tissues of light and heavy rabbits, and there was a significant difference compared to other tissues (Figure 1D). These data suggested that miR-383-5p may be related to adipose development.

### 2.2. Analysis of Differentially Expressed Genes

To further understand the function of miR-383-5p, we sent the miR-383-5p mimic group and miR-383-5p NC group to the company for transcriptome sequencing. Six cDNA libraries were constructed from preadipocytes. As shown in Appendix A, 262508156 clean readings were obtained. The levels of Q20 in the miR-383-5p mimic group and the NC group were equivalent (percentage of readings with Phred quality value > 20), ranging from 97.18% to 97.62%. The GC content of libraries ranged from 52.73% to 53.91%, with an average content of 53.31%. The comparative analysis showed that the average comparison rate of samples was 89.39%. Therefore, all libraries were of high quality and could be used for further analysis. String Tie was used to assemble new transcripts, and Pfam, SUPERFAMILY, GO, and KEGG were used to annotate them. A total of 64 unknown genes, 39 novel lncRNAs and 1514 novel protein-coding genes were obtained. A total of 1642 (721 up-regulated and 921 down-regulated) known differentially expressed genes were screened from the sequencing library using DESeq2 (Appendix A). Values of log2 (fold change) and -log10 (*p*-value) were used to construct volcano figures for differentially expressed genes (Figure 2A). Three differentially expressed genes were randomly selected for qRT-PCR validation, and the results showed a similar trend to that of the sequencing (Figure 2B). In addition, we used the ENCORI website to identify 102 candidate genes out of 921 down-regulated genes that have potential binding sites with miR-383-5p (Figure 2C). Then, three highly expressed genes were selected from 102 candidate genes for qRT-PCR validation, including *NUF2*, *CNIH4*, and *RAD51AP1*. The results were consistent with the sequencing results (Figure 2D). Finally, the GO analysis results showed that 763 GO items were enriched (400 biological processes (BP), 107 cell composition (CC), and 256 molecular functions (MF)) (Figure 2E) (Appendix A). The main biological processes involved protein catabolism, organic nitrogen compound biosynthesis, and lipid biosynthesis, especially some GO terms related to glucose metabolism. KEGG analysis showed that a total of 305 pathways were enriched, among which oxidative phosphorylation, non-alcoholic fatty liver disease and NOD receptor signal pathways were significantly enriched (Figure 2F) (Appendix A).

### 2.3. miR-383-5p Promotes Rabbit Preadipocyte Proliferation

To explore the role of miR-383-5p in the proliferation of rabbit preadipocytes, miR-383-5p mimic, miR-383-5p inhibitor, miR-383-5p negative control (NC) and miR-383-5p inhibitor negative control (INC) were transfected into cells. As shown in Figure 3A, the relative expression levels of miR-383-5p in the mimic (inhibitor) group were significantly higher (lower) than those in the NC (INC) group, indicating that transfection analogs successfully increased (decreased) the expression levels of miR-383-5p. As shown in Figure 3B,C, the CCK-8 results showed a significant increase in absorbance in the miR-383-5p mimic group at 0, 24, 48 and 72 h after transfection, while the miR-383-5p inhibition group showed the opposite trend. The positive cell rate in the miR-383-5p mimic group was significantly higher than that in the NC group. The positive cell rate in the miR-383-5p inhibitor group was significantly lower than that in the INC group (Figure 3D,E). Moreover, the results of WB showed that compared with the NC group, the expression of CDK2 and CDK4 in the miR-383-5p mimic group increased, and the expression in the miR-383-5p inhibitor group decreased (Figure 3F,G) (Appendix A). The comprehensive results showed that miR-383-5p has a promotional effect on the proliferation of rabbit preadipocytes. 

### 2.4. miR-383-5p Promotes Rabbit Preadipocyte Differentiation

To investigate the function of miR-383-5p in the differentiation of rabbit preadipocytes, the fourth generation of preadipocytes were cultured in cell culture plates and differentiated with the above method when the density reached 80%. We first conducted an Oil Red O staining experiment; the results showed that lipid droplets rapidly increased during the differentiation of preadipocytes (Figure 4A). Meanwhile, the expression of miR-383-5p reached its peak on 6 d, revealing a possible regulatory role of miR-383-5p in preadipocyte differentiation (Figure 4B). Afterwards, Oil Red O staining showed that the lipid deposition was higher in the mimic group than the NC group, but the lipid deposition was lower in the inhibitor group than the INC group after 6 d of transfection miR-383-5p mimic, miR-383-5p inhibitor, NC, and INC (Figure 4C,D). The qRT-PCR results showed an obvious and highly significant increase of genes in the mimic group, including *PPARγ*, *FABP4*, and *CEBP/α*; in contrast, these genes were expressed at lower levels in the inhibitor group (Figure 4E,F). As shown in Figure 4G,H, the results of WB showed that, compared with the NC group, the expression of SREBP1 and FABP4 in the miR-383-5p mimic group increased, and the expression in the miR-383-5p inhibitor group decreased (Appendix A). Therefore, we concluded that miR-383-5p plays a positive role in rabbit preadipocyte differentiation.

### 2.5. Effects of miR-383-5p on Rabbit Preadipocytes Treated with Palmitic Acid

In order to explore the effect of miR-383-5p on palmitic acid (PA)-treated rabbit preadipocytes, we first constructed a model of PA-treated rabbit preadipocytes. After Oil Red O staining and triglyceride determination, we found that the lipid deposition in the PA group was significantly higher than in the blank group (CON) group (Figure 5A,B). The qRT-PCR results showed a significant increase in the expression of miR-383-5p in the PA group (Figure 5C). Moreover, Oil Red O staining showed that the lipid deposition was higher in the miR-383-5p mimic group than in the NC group, but the lipid deposition was lower in the miR-383-5p inhibitor group than in the INC group after 8 d of transfection (Figure 5D,E). TG results showed that the miR-383-5p mimic group had higher levels than the NC group and the miR-383-5p inhibitor group had lower levels than the INC group (Figure 5F). The expression of hypertrophy marker genes *ACC*, *FAS* and *SCD* were detected by qRT-PCR. *ACC*, *FAS*, and *SCD* were significantly increased in the miR-383-5p mimic group but decreased in the miR-383-5p inhibitor group (Figure 5H). These data suggested that miR-383-5p plays a positive role in rabbit preadipocytes treated with PA.

### 2.6. miR-383-5p Regulates the Proliferation and Differentiation of Rabbit Preadipocytes by Targeting RAD51AP1

To further explore the potential underlying mechanism of miR-383-5p regulation of adipogenesis, miRanda was used to predict potential target genes of miR-383-5p. Among the 921 downregulated genes, there are a total of 102 candidate genes, including the 3′UTR of *RAD51AP1* (Figure 6A). The results of qRT-PCR experiments showed that *RAD51AP1* was significantly down-regulated in the miR-383-5p mimic group (Figure 2C). The WB experiment verified the result (Figure 6B). The luciferase reporter assay revealed that the luciferase activity was highly suppressed in the group containing the wild-type 3′UTR of *RAD51AP1* mRNA but not significantly changed in the mutant group (Figure 6C). These observations imply that *RAD51AP1* was a direct miR-383-5p target. To reveal the function of *RAD51AP1* on rabbit preadipocytes, si-RAD51AP1 and si-RAD51AP1-NC were transfected into cells. As shown in Figure 6D,E, the WB results showed that in the si-RAD51AP1-NC group, the expression of PCNA and CDK2 increased at the protein levels but decreased in the si-RAD51AP1 group. Moreover, the EDU proliferation assay revealed that the number of positive cells was significantly higher in the si-RAD51AP1-NC group than the si-RAD51AP1 group. (Figure 6F,G). These data suggested that si-RAD51AP1 played an inhibitory role in rabbit preadipocyte proliferation. Oil Red O staining showed that the lipid deposition was higher in the si-RAD51AP1 group than in the si-RAD51AP1-NC group after 6 d of transfection (Figure 6H,I). The WB results showed that in the si-RAD51AP1 group, the expression of PPARγ and C/EBPα increased at the protein level (Figure 6J,K). These data suggested that si-RAD51AP1 played a positive role in rabbit preadipocyte differentiation.

Finally, we also investigated whether miR-383-5p can regulate the proliferation and differentiation of rabbit preadipocytes by targeting *RAD51AP1*. miR-383-5p INC/si-RAD51AP1-NC, miR-383-5p inhibitor/si-RAD51AP1-NC, miR-383-5p INC/si-RAD51AP1, and miR-383-5p inhibitor/si-RAD51AP1 were co-transfected into rabbit preadipocytes, respectively. The EDU proliferation assay revealed that the number of positive cells was significantly higher in the miR-383-5p inhibitor/si-RAD51AP1 group than in the miR-383-5p inhibitor/si-RAD51AP1-NC and miR-383-5p INC/si-RAD51AP1 group (Figure 6L,M). As shown in Figure 6N, the CCK-8 results showed a significant increase in absorbance in the miR-383-5p inhibitor/si-RAD51AP1 group at 0, 24, 48 and 72 h after transfection. Oil Red O staining showed that the lipid deposition was higher in the miR-383-5p INC/si-RAD51AP1 group than in the miR-383-5p inhibitor/si-RAD51AP1 group, but the lipid deposition was lower in the miR-383-5p inhibitor/si-RAD51AP1-NC group than in the miR-383-5p inhibitor/si-RAD51AP1 after 6 d of transfection (Figure 6O,P). The WB results showed the same trend (Figure 6Q,R) (Appendix A). The above results indicate that miR-383-5p may play a potential role in adipogenesis and development by targeting *RAD51AP1.*

## 3. Discussion

In higher vertebrates, adipose tissue is an organ that performs a lot of significant physiological functions [25]. Excess adipose tissue in the body can cause many organs and systems to be in a pathological state [26]. Adipocytes can undergo a transformation and change their structure and metabolism. The transition is driven by the concerted control of endogenous regulators, including miRNAs [27]. With the improvement in people’s living standards, more and more people are becoming obese, and this has reached epidemic level [28]. However, the prevention and treatment of obesity has not yet been successful. Here, we found that miR-383-5p is highly expressed in the perirenal fat of high-fat-fed rabbits, but it is not yet known whether miR-383-5p is involved in lipid metabolism. Subsequently, we conduct a detailed study of the proliferation and differentiation of rabbit preadipocytes using miR-383-5p to better understand the importance of miRNAs in lipid metabolism and to attempt to reveal the molecular regulatory mechanisms related to miRNAs in obesity treatment.

The same miRNA has different expressions in different tissues, and many miRNAs also have different expressions in adipose tissue, often with co-expression [29]. In our study, we found that miR-383-5p_1 had a high expression level in adipose tissue, which was significantly different from that in other tissues, and the expression level in heavy rabbits was significantly higher than that in light rabbits. These data revealed that miR-383-5p may play an important role in lipid metabolism. In adipose tissue, both increased adipocyte size and/or increased adipocyte number characterize the increased fat mass associated with obesity. The number of adipocytes largely determines the process of adipogenesis in body fat depots [30]. The proliferation of adipocyte precursors contributes to the development of obesity in mammals [31]. Therefore, we studied the effect of miR-383-5p on the proliferation and differentiation of rabbit preadipocytes. At the stage of preadipocyte proliferation, miR-383-5p plays a positive role. We found that CDK2 and CDK4 proteins were highly expressed in the miR-383-5p mimic group. It is worth noting that cyclin-dependent kinase 2 (*CDK2*) and *CDK4* are key cell-cycle regulators [32]. *CDK2* can phosphorylate many substrates to drive progression through the cell cycle [33]. Moreover, *CDK4* deficiency has been found to affect cell proliferation in human malignant tumor cells [34]. At the stage of preadipocyte differentiation, miR-383-5p plays a positive role in preadipocyte differentiation. *PPARγ*, *FABP4*, *C/EBPα*, and *SREBP1* are widely accepted as the essential transcription factors of preadipocyte differentiation [35]. The upregulation of miR-383-5p increased the expression of adipogenic marker genes *FABP4*, *PPARγ*, and *C/EBPα*, whereas the downregulation of miR-383-5p using the synthetic inhibitor markedly reduced the formation of neutral lipid droplets and suppressed the expression of marker genes at both the *FABP4*, *PPARγ*, and *C/EBPα* mRNAs and protein levels.

Research has shown that hyperplasia of adipose tissue occurs only in certain periods of life [36]. Adipocyte hypertrophy is the main process connected with obesity in adults [37]. The concentration of TG in cells is positively correlated with the increase in adipocyte area, which can better reflect the degree of adipocyte hypertrophy [30]. PA was chosen as our stimulus because it is the most common saturated fatty acid in the human body and is elevated in individuals with obesity due to lipolysis [38]. Secondly, PA is the physiological component of TG in adipocytes [39]. Finally, PA is noteworthy for its strong effects on gene expression, and can induce the differentiation of preadipocytes [40,41]. In the current study, we cultured rabbit preadipocytes with conditional medium derived from differentiated adipocytes incubated with PA. *ACC*, *FAS*, and *SCD* are widely accepted as the specific genes for fatty acid synthesis [42]. The upregulation of miR-383-5p increased the expression of marker genes *ACC*, *FAS*, and *SCD*, whereas the downregulation of miR-383-5p using the synthetic inhibitor markedly reduced the formation of neutral lipid droplets and suppressed the expression of marker genes at both the *SCD* mRNA and protein levels. 

A total of 1642 known differentially expressed genes were screened from the sequencing library using DESeq2. GO and KEGG enriched the pathways related to lipid metabolism. Therefore, miR-383-5p may be important regulators in adipogenesis. In addition, bioinformatic predictions show that there are 102 candidate genes for miR-383-5p, including *CNIH4*, *NUF2*, and *RAD51AP1*. *CNIH4* is a novel biomarker associated with poor prognosis and cell proliferation in low-grade glioma patients [43]. *NUF2* plays an important role in tumorigenesis and is upregulated in multiple human cancers, including serous adenocarcinoma, liver cancer, and colorectal cancer [44]. Moreover, a large number of studies have shown that *RAD51AP1* plays a key role in the development of cancer cells. Allison et al. [45] unraveled the role of *RAD51AP1*, a protein involved in homologous recombination in genotoxic carcinogen (azoxymethane, AOM)-induced colorectal cancer. *RAD51AP1* might accelerate the progression of ovarian cancer by the TGF-β/Smad signaling pathway [46]. *RAD51AP1* silencing significantly inhibited cell proliferation and invasion in esophageal squamous cell carcinoma (ESCC), thereby highlighting its potential as a novel target for ESCC treatment [47]. However, there is limited research on the regulation of adipogenesis by *RAD51AP1*. Among this research, *RAD51AP1* was found in prenatal testosterone-induced visceral adipose tissue of female sheep and was highly expressed in the adipose-derived stem cells, which may be involved in lipid metabolism processes [48,49]. In our research, *RAD51AP1* was a direct miR-383-5p target. *RAD51AP1* may provide a new therapeutic target for metabolic diseases caused by obesity, which can help us better understand fat production and metabolism.

## 4. Materials and Methods

### 4.1. Ethics Statement

All experiments in the present study involving animals were performed under the direction of the Institutional Animal Care and Use Committee from the College of Animal Science and Technology, Sichuan Agricultural University, China. (Certification No. SYXK2019-187).

### 4.2. Animals and Cell Collection

A newly born Tianfu black rabbit was used for sampling. The experimental rabbit was rubbed all over with alcohol and sacrificed humanely to reduce suffering. The primary preadipocytes were collected in the perirenal area of Tianfu black rabbits using the method described earlier [50]. The tissue samples of 35-day-old rabbits were obtained from experiments conducted by Wang et al. [24]. Briefly, 24 female Tianfu black rabbits were raised in the rabbit farm of Sichuan Agricultural University and randomly divided into two groups. Twelve rabbits were fed a standard diet (SON) (light rabbits), twelve rabbits were fed a high-fat diet (HFD: 10% lard was added to SON) (heavy rabbits); the feed was supplied three times a day, and free drinking water was used. After four weeks, the rabbits were slaughtered, and tissue samples were collected. At the same time, body weight, TG, TC, and body fat rate were measured as indicators to distinguish between light and heavy rabbits.

### 4.3. Bioinformatic Analysis of Sequencing Data

The sample RNA was prepared, and strict quality control was carried out on the RNA sample, mainly through the Agilent 2100 bioanalyzer (Agilent technologies, Santa Clara, CA, USA). The first cDNA strand was synthesized in M-MuLV reverse transcriptase system, the second cDNA strand was synthesized in dNTPs and DNA polymerase I, poly (A) tails were added, and sequencing connectors were connected to generate 250–300 BP cDNA, and PCR-amplified cDNA to build the cDNA libraries using the NEBNext^®^ UltraTM RNA Library Prep Kit for Illumina^®^ (New England Biolabs, Ipswich, MA, USA). After passing the library inspection, the different libraries were pooled according to the effective concentration and target offline-data volume requirements, and then Illumina sequencing was performed. Differential expression analysis was performed using the DESeq2 with *p* ≤ 0.05.

### 4.4. Establish the Hypertrophic Cell Model

A total of 4 mmol PA (Sigma, Shanghai, China) was added to 0.1 mol/L NaOH (50 mL) for 30 min at 70 °C and then mixed with 10% bovine serum albumin (BSA, Sigma, Shanghai, China) to obtain the 1 mmol/L PA mother liquor. The mother liquor was diluted with the serum-free medium and filtered to obtain a 0.3 mmol/L PA working solution. For the CON group, the mother liquor was replaced with the same volume of 10% BSA and mixed with the serum-free medium. After transfection, PA and CON reagents were added, and samples were collected 2 d later. 

### 4.5. RNA Extraction and Quantitative Real-Time PCR (qRT-PCR)

Total RNA from the samples was extracted using the Total RNA Extraction Kit (Solarbio, Beijing, China), following the guidelines of the manufacturer. Reverse transcription of mRNA and miRNA was performed using the RT Easy^TM^ II (With gDNase) (FOREGENE, Chengdu, China) and the Mir-X^TM^ miRNA First-Strand Synthesis Kit (Takara, Dalian, China), respectively. Then, qRT-PCR was performed in triplicate using the NovoStart^®^SYBR qPCR SuperMix plus (Novoprotein, Shanghai, China) on a CFX96 instrument (Bio-Rad, Hercules, CA, USA), and the relative levels of mRNAs and miRNAs were calculated using the 2^−ΔΔCt^ method. U6 and GAPDH were used as the internal reference for mRNA quantification. The sequences of primers are shown in Appendix A and were synthesized by Sangon Biotech (Shanghai, China). 

### 4.6. Cell Culture and Induced Differentiation

Preadipocytes were maintained in a growth medium (GM) containing 10% fetal bovine serum (FBS, Gibco, CA, USA) in an incubator (Thermo Fisher Scientific, San Jose, CA, USA) at 37 °C and 5% CO_2_ environment, and the medium was changed every 24 h. When the cell density reached 70–80%, the induced differentiation medium (5% FBS, 1 μmol/L dexamethasone (DEX; Solarbio, Beijing, China), 0.5 mmol/L 3-isobutyl-1-methylxanthine (IBMX; Solarbio, Beijing, China) and 10 ug/mL insulin (Solarbio, Beijing, China)) was used to induce differentiation for 3 d to observe the lipid droplet formation of preadipocytes. Then, the medium was replaced with a maintenance medium containing 5% FBS and 10 ug/mL insulin for 3 d. Subsequently, the medium was replaced with GM to obtain mature adipocytes.

### 4.7. Transfection

In the proliferation assay, miR-383-5p mimic, miR-383-5p inhibitor, NC, INC, si-RAD51AP1, and si-RAD51AP1 were transfected using lipoMax (Sudgen, Nanjing, China) when the cell density reached 50–60%. After 6 h, the medium was changed to GM. In the differentiation assay, when the cell density reached 80% LipoMax was used for transfection. After 6 h, the medium was changed to the induction differentiation medium. RNA oligo was synthesized by Sangon Biotech (Shanghai, China). The miRNA mimic is chemically synthesized mature double-stranded miRNA that enhances endogenous miRNA function. This includes a sequence that is consistent with the target miRNA mature body sequence and a sequence that is complementary to the miRNA mature body sequence. The miRNA inhibitor is a chemically synthesized mature complementary single-stranded miRNA designed with methoxy modification. The inhibitor specifically targeting miRNA in cells can efficiently inhibit the activity of endogenous miRNA in organisms and conduct miRNA functional deficiency research. NC and INC are negative controls, respectively. The detailed RNA oligo sequences are shown in Appendix A.

### 4.8. Measurement of Triglycerides and Cholesterol

The blood sample was centrifuged at 4 °C for 5 min and the serum was transferred to a clean pipe. According to the standard guides, the serum concentration of TG and cholesterol TC were respectively tested using the commercial kit based on enzymatic colorimetry (Jiancheng, Nanjing, China). Similarly, preadipocytes were inoculated in the 6-well plates (NEST Biotechnology, Wuxi, China), and transfected with the miR-383-5p mimic, NC, miR-383-5p inhibitor, and INC when the cell density was about 70%. After inducing differentiation, cells were collected, broken up, and tested for TG content with the above kit.

### 4.9. Western Blotting

Total protein from the cell was collected using the ProteinExt^®^Mammalian Total Protein Extraction Kit (TransGen Biotech, Beijing, China), following the manufacturer’s protocol. The concentration of the protein was measured using the Bradford protein assay kit (Novoprotein, Shanghai, China). The protein was resolved on 10% SDS-PAGE and then transferred to a PVDF membrane, followed by sealing of the sealing fluid. The membranes were incubated with the corresponding primary antibodies for 8 h at 4 °C and subsequently incubated with the secondary anti-bodies (Goat Anti-Rabbit IgG H&L (HRP), Zen bioscience, Chengdu, China) for 2 h. Then the bands were washed three times, and the hypersensitive chromogenic solution was developed using chemiluminescence in Touch Imager Pro, e-BLOT Life Science (Shanghai) Co., Ltd. (Shanghai, China). The primary antibodies PCNA, CDK2, and CDK4 were purchased from Zen bioscience (Chengdu, China). The primary antibodies PPARγ, CEBP/α, FABP4, SREBP1, and RAD51 were purchased from the Abclonal (Wuhan, China).

### 4.10. Cell Counting Kit 8 (CCK-8) Assay

Preadipocytes were inoculated in the 96-well plates (NEST Biotechnology, Wuxi, China) and transfected with the miR-383-5p mimic, NC, miR-383-5p inhibitor, and INC when the cell density was about 60%. According to manufacturer’s instructions, 10 μL CCK-8 was added to each well at 0, 24, 48 and 72 h after transfection, and the cells were cultured at 37 °C for 2 h in 5% CO_2_ incubator. The absorbance at 450 nm was measured by Thermo Scientific^TM^ Varioskan LUX (Thermo Scientific, Waltham, MA, USA). The absorbance was drawn and analyzed using GraphPad Prism 9 (GraphPad Software Inc., La Jolla, CA, USA).

### 4.11. EDU Proliferation Assay

Preadipocytes were inoculated in 24-well plates. After 24 h of transfection, the cells were cultured in a complete medium containing 50 μL 5-ethynyl-2′-deoxyuridine (EDU, RiboBio, Guangzhou, China) for 2 h and fixed and dyed according to manufacturer’s instructions. The nucleus and positive cells were stained with Hoechst and EDU, respectively, and photographed using an inverted fluorescence microscope (Olympus, Tokyo, Japan) at the same visual field conditions. Images were analyzed using image-pro plus 6.0 software (Media Cybernetics, Inc., Rockville, MD, USA).

### 4.12. Oil Red O Staining

Preadipocytes were inoculated in the 35 mm cell culture dishes (NEST Biotechnology, Wuxi, China). The cells were washed two times with phosphate-buffered saline (PBS) and fixed in 10% paraformaldehyde for 30 min. Subsequently, Oil Red O (NJBI, Nanjing, China) was mixed with deionized water at a rate of 3:2 and then added to the stained cell for 25 min. Finally, the cells were washed with PBS. Images were collected under an inverted microscope. Then, 2 mL of isopropyl alcohol was added into the cell culture dishes, and the cells were fully lysed and then transferred into the 96-well plate. The absorbance at 510 nm was measured by Thermo Scientific^TM^ Varioskan LUX. The absorbance was drawn and analyzed using GraphPad Prism 9 (GraphPad Software Inc., La Jolla, CA, USA). Moreover, the count of lipid droplets was measured using image-pro plus 6.0 software (Media Cybernetics).

### 4.13. Dual-Luciferase Reporter Assay

The MiRanda database (https://www.microRNA.org; accessed on 1 March 2022) was used to predict miR-383-5p-related target genes. The potential binding site between miRNA and mRNA was predicted using the RNA22 website (https://cm.jefferson.edu/rna22/Interactive/; accessed on 15 January 2023), based on the sequence information. According to the results, *RAD51AP1* was selected as the possible target gene of miR-383-5p. Luciferase reporter plasmids (wild-type (WT) and mutant (MUT) of target sequence) were constructed by Tsingke Biotechnology Co., Ltd. (Beijing, China). Next, 293T cells were seeded into 24-well plates (NEST Biotechnology, Wuxi, China). The miR-383-5p mimic or NC was co-transfected with specific WT or mutant plasmids using the lipofectamine 3000 reagent (Invitrogen, Carlsbad, CA, USA) when the cell density reached 70%. Then, luciferase activities were measured using the Duo-Lite TM Luciferase Assay System (Vazyme, Nanjing, China) after Aa24 h.

### 4.14. Statistical Analysis

All data were analyzed using SPSS 20.0 statistical software; they were consistent with normal distribution and presented as means ± SEM. Student’s *t*-test and one-way analyses were used to analyze the significance of differences between the groups (* *p* < 0.05; ** *p* < 0.01).

## 5. Conclusions

Our research results indicate that miR-383-5p can promote the proliferation and differentiation of rabbit preadipocytes, and there is a direct targeting relationship with *RAD51AP1*. This reveals a new molecular mechanism of miR-383-5p regulating adipogenesis, which can provide theoretical guidance and new ideas for the treatment of metabolic diseases such as obesity. 

## Figures and Tables

**Figure 1 ijms-24-14025-f001:**
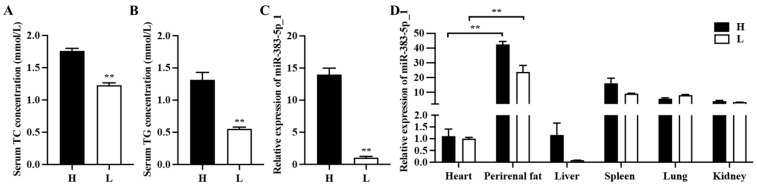
miR-383-5p_1 is associated with adipose development. (**A**,**B**) TG and TC content in light (L) and heavy (H) rabbits. (**C**) The expression of miR-383-5p_1 in perirenal adipose tissue of light and heavy rabbits. (**D**) Expression of miR-383-5p_1 in light and heavy rabbit tissues. Data are expressed as means ± SEM. ** *p* < 0.01.

**Figure 2 ijms-24-14025-f002:**
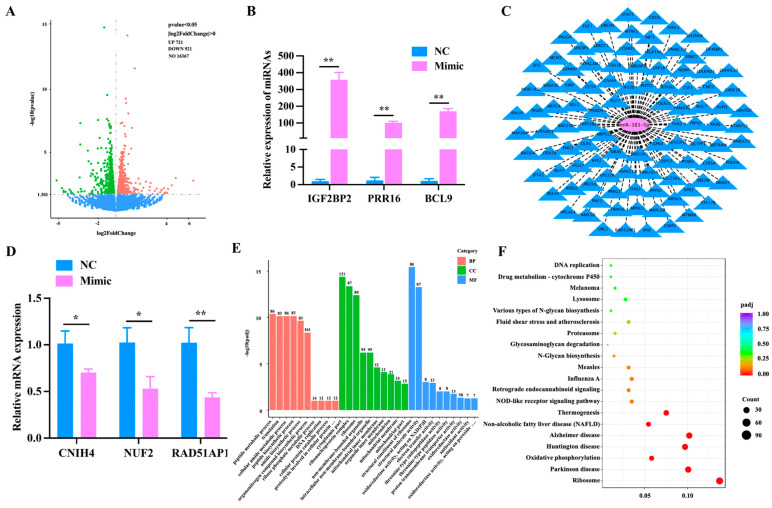
Analysis of differentially expressed genes. (**A**) The volcano plot was constructed for the differentially expressed genes based on log2 (fold change) and −log10 (*p*-value). (**B**) Validation of differentially expressed genes. (**C**) The intersection target genes of miR-383-5p predicted by miRanda, TargetScan, and RNAhybrid software. (**D**) *CNIH4*, *NUF2*, and *RAD51AP1* levels after transfecting with the miR-383-5p mimic and NC 24 h. (**E**) GO analysis of the differentially expressed genes were performed, imaging only those terms that were significantly enriched in the BP, CC, and MF categories. (F) KEGG analysis of the differentially expressed genes only listed significant enrichment pathways. The data are presented as means ± SEM. * *p* < 0.05; ** *p* < 0.01.

**Figure 3 ijms-24-14025-f003:**
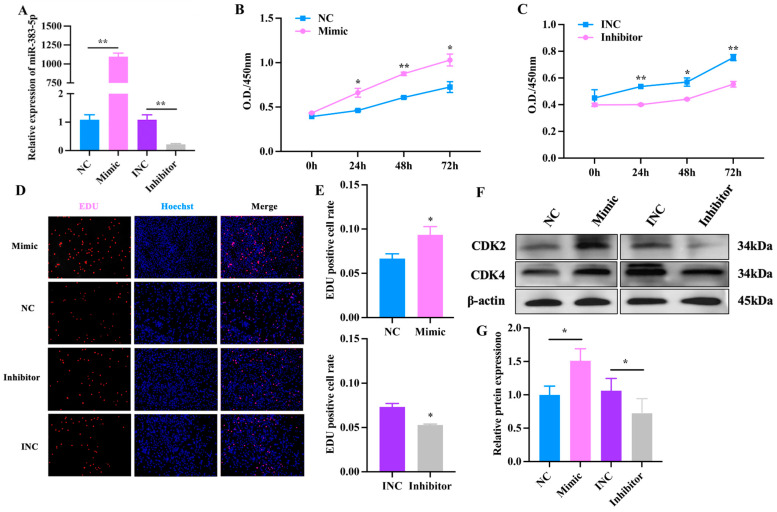
miR-383-5p promotes rabbit preadipocyte proliferation. (**A**) Transfection efficiency detection of miR-383-5p mimic and inhibitor (*n* = 9).(**B**,**C**) The absorbance of preadipocytes at 0, 24, 48, and 72 h after transfection with the miR-383-5p mimic, NC, miR-383-5p inhibitor, and INC (*n* = 6). (**D**) The picture of the EDU proliferation assay for preadipocytes transfected with the miR-383-5p mimic, NC, miR-383-5p inhibitor, and INC. Red fluorescence represents the EDU-positive cells, and blue fluorescence represents the Hoechst-stained cells. (**E**) The percent of EDU-positive cells. EDUpositive cells rate = EDU-positive cells/Hoechst-stained cells × 100% (*n* = 3). (**F**,**G**) CDK2 and CDK4 protein levels during preadipocyte proliferation after transfecting with NC, miR-383-5p mimic, INC, and miR-383-5p inhibitor (*n* = 3). The data are presented as means ± SEM. * *p* < 0.05; ** *p* < 0.01.

**Figure 4 ijms-24-14025-f004:**
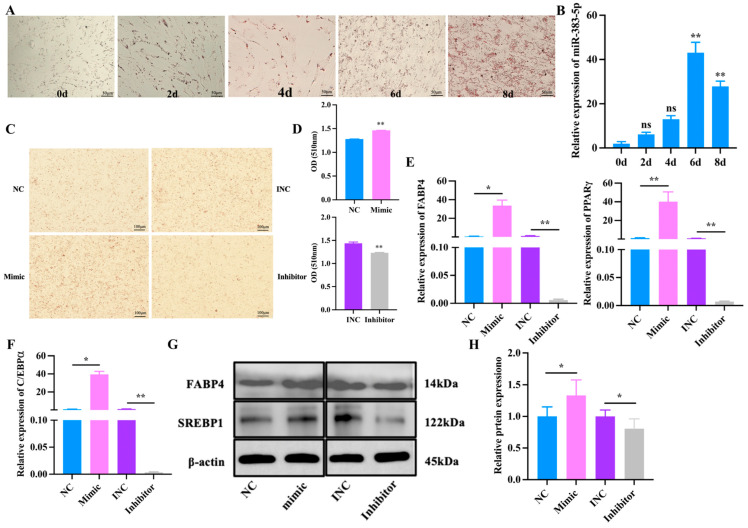
miR-383-5p promotes rabbit preadipocyte differentiation. (**A**) Oil Red O staining of lipid droplets at 0, 2, 4, 6, and 8 d of differentiation. (**B**) The relative expression levels of miR-383-5p during preadipocyte differentiation progress (*n* = 9). (**C**,**D**) Oil Red O staining of lipid droplets after 6 d of transfection and quantitative results of Oil Red O staining (*n* = 9). (**E**,**F**) The relative expression levels of *PPARγ*, *FABP4*, and *C/EBPα* in rabbit preadipocytes induced differentiation at 6 d after transfecting with NC, miR-383-5p mimic, INC, and miR-383-5p inhibitor (*n* = 9). (**G**,**H**) SREBP1 and FABP4 protein levels during preadipocyte differentiation after transfecting with NC, miR-383-5p mimic, INC, and miR-383-5p inhibitor (*n* = 3). The data are presented as means ± SEM. * *p* < 0.05; ** *p* < 0.01.

**Figure 5 ijms-24-14025-f005:**
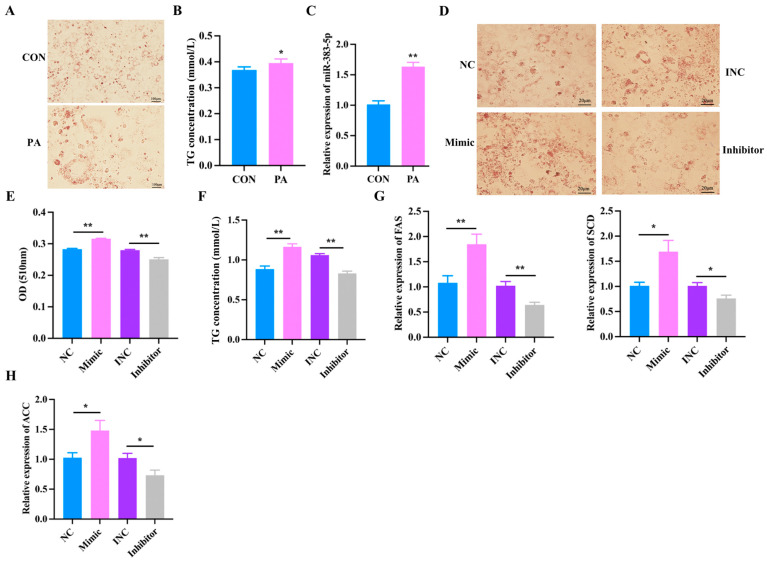
Effects of miR-383-5p on rabbit preadipocytes treated with palmitic acid. (**A**) Oil Red O staining of lipid droplets after 2 d of PA treatment. (**B**) TG content after 2 d of PA treatment. (**C**) Relative expression of miR-383-5 after 2 d of PA treatment. (**D**,**E**) Oil Red O staining and quantitative results of lipid droplets 8 d after transfection. (**F**) TG content in rabbit preadipocytes induced differentiation at 6 d after transfecting and 2 d after treatment with PA (*n* = 3). (**G**,**H**) The relative expression levels of *ACC*, *FAS*, and *SCD* in rabbit preadipocytes induced differentiation at 6 d after transfecting with NC, miR-383-5p mimic, INC, and miR-383-5p inhibitor, and 2 d after treatment with PA (*n* = 9). The data are presented as means ± SEM. * *p* < 0.05; ** *p* < 0.01.

**Figure 6 ijms-24-14025-f006:**
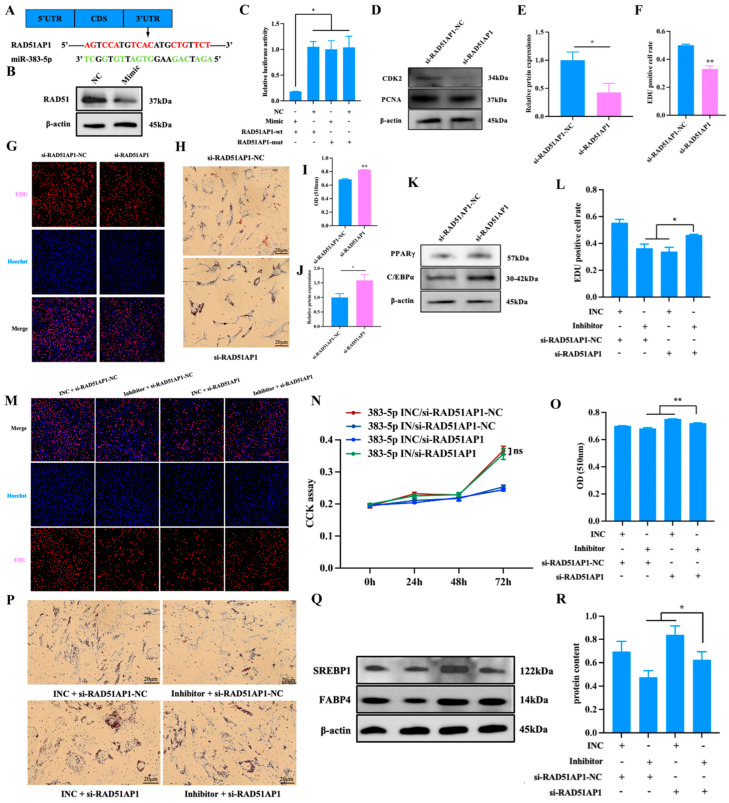
miR-383-5p regulates the proliferation and differentiation of rabbit preadipocytes by targeting *RAD51AP1*. (**A**) The predicted binding site of gene *RAD51AP1* with miR-383-5p. (**B**) After transfection with miR-383-5p mimic and NC, the expression of RAD51 was determined by WB. (**C**) Luciferase assays were performed by co-transfection of *RAD51AP1* WT and mutant plasmids with miR-383-5p mimic and NC, respectively, in 293T cells, and the WT + NC group was used as the control group (*n* = 3). (**D**,**E**) CDK2 and PCNA protein levels during preadipocyte proliferation after transfecting with si-RAD51AP1 and si-RAD51AP1-NC (*n* = 3). (**F**) The percent of EDU-positive cells. EDU-positive cell rate = EDU-positive cells/Hoechst-stained cells × 100% (*n* = 3). (**G**) The picture of the EDU proliferation assay for preadipocytes transfected with the si-RAD51AP1 and si-RAD51AP1-NC. Red fluorescence represents the EDU-positive cells, and blue fluorescence represents the Hoechst-stained cells. (**H**,**I**) Oil Red O staining and quantitative results of lipid droplets 6 d after transfection. (**J**,**K**) PPARγ and C/EBPα protein levels during preadipocytes differentiation after transfecting with si-RAD51AP1 and si-RAD51AP1-NC (*n* = 3). (**L**) The percent of EDU-positive cells. EDU-positive cell rate = EDU-positive cells/Hoechst-stained cells × 100% (*n* = 3). (**M**) The picture of the EDU proliferation assay for preadipocytes transfected with the miR-383-5p INC/si-RAD51AP1-NC, miR-383-5p inhibitor/si-RAD51AP1-NC, miR-383-5p INC/si-RAD51AP1, and miR-383-5p inhibitor/si-RAD51AP1. Red fluorescence represents the EDU-positive cells, and blue fluorescence represents the Hoechst-stained cells. (**N**) The absorbance of preadipocytes at 0, 24, 48, and 72 h after transfection with the miR-383-5p INC/si-RAD51AP1-NC, miR-383-5p inhibitor/si-RAD51AP1-NC, miR-383-5p INC/si-RAD51AP1, and miR-383-5p inhibitor/si-RAD51AP1 (*n* = 6). (**O**) Quantitative results of Oil Red O staining (*n* = 5). (**P**) Oil Red O staining of lipid droplets after 6 d of transfection. (**Q**,**R**) SREBP1 and FABP4 protein levels during preadipocyte differentiation after transfecting with miR-383-5p INC/si-RAD51AP1-NC, miR-383-5p inhibitor/si-RAD51AP1-NC, miR-383-5p INC/si-RAD51AP1, and miR-383-5p inhibitor/si-RAD51AP1 (*n* = 3). The data were presented as means ± SEM. * *p* < 0.05; ** *p* < 0.01.

## Data Availability

The raw data is publicly available on NCBI portal at Sequence Read Archive (SRA) BioProject ID: PRJNA904507. RNA-Seq data Submission ID: SUB12315011.

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
