# Peer review of "miR-383-5p Regulates Preadipocyte Proliferation and Differentiation by Targeting RAD51AP1"

_ijms, 2023, doi:10.3390/ijms241814025_

Round 1

Reviewer 1 Report (Previous Reviewer 1)

Comments and Suggestions for Authors

The manuscript by Wang et al addresses association of differential expression by miRNA regulation in adipocytes cells, as study design to know the lipid metabolism. Increasing evidence suggests that miR-383-5p controls the cell proliferation and cell differentiation, as features of functions of adipocytes, and target gene regulation; by using cell culture of primary cell, RNA-sequencing, real-time PCR, functional analysis and luciferase assays; highlighting that high and low miR-383-5p levels are associated with proliferation and differentiation. Here, the authors investigate the regulation of miR-383-5p linked to the regulation of lipid metabolism based on animal model.

Material and Methods                                              

In section 4.2.- The authors should describe the feeding protocol for the rabbits and the biochemical parameters between groups of rabbits (heavy and light).

Results

1.- In Section 2.3 .- The Figure 3E should be complemented with graph in bar or box plot (Protein concentration).

2.- In Section 2.4 .- The Figure 4E should be complemented with graph in bar or box plot (Protein concentration).                                                                                                                               

4.- In Section 2.6 .- The authors study the RAD51AP1 regulation by miR-383-5p using luciferase reporter assay. The authors indicate the selection of RAD51AP1 according to Figure 2C and Figure 6B instead of CNIH4 and NUF2. The authors should show statistically significant changes of RAD51AP1 compared to CNIH4 and NUF2.

Author Response

Dear Reviewer,

Thanks very much for taking your time to review this manuscript. I really appreciate all your comments and suggestions! Please find my itemized reply in the document.

Reviewer 2 Report (New Reviewer)

Comments and Suggestions for Authors

Author Response

Dear Reviewer,

Thanks very much for taking your time to review this manuscript. I really appreciate all your comments and suggestions! Please find my itemized reply in the document.

Round 2

Reviewer 2 Report (New Reviewer)

Comments and Suggestions for Authors

The authors made significant changes in the revised manuscript and it can be accepted in its present form.

This manuscript is a resubmission of an earlier submission. The following is a list of the peer review reports and author responses from that submission.

Round 1

Reviewer 1 Report

Comments and Suggestions for Authors

The manuscript by Wang et al addresses association of differential expression by miRNA regulation in adipocytes cells, as study design to know the lipid metabolism. Increasing evidence suggests that miR-383-5p controls the cell proliferation and cell differentiation, as features of functions of adipocytes, and target gene regulation; by using cell culture of primary cell, RNA-sequencing, real-time PCR, functional analysis and luciferase assays; highlighting that high and low miR-383-5p levels are associated with proliferation and differentiation. Here, the authors investigate the regulation of miR-383-5p linked to the regulation of lipid metabolism based on animal model.

Major concerns

1.- The authors study the association between miR-383-5p and features of functions in adipocytes from perirenal area of rabbits. However, there are some concerns that should be explained by the authors.

According to results in Section 3.1, the authors analysis the differential levels of TC, TG and miR-383-5p in group (H) vs (L). However, in Figure 1D, the results for Group (L) are not shown. This manuscript is a comparative study between two groups (H and L) and differential expression and functional analysis considering these group should be shown in all findings.

Could these findings be found in the group (L)?

In section 3.2, the 721 genes up-regulated were not studied by the authors. What was the criterion to choose 102 candidates from 921 down-regulated genes. The authors stated that NUF2, CNIH4 and RAD51AP1 were randomly selected for qRT-PCR validation. The authors should explain a valid criterion of selection.

2. - To better understand and improve the relevance of the present manuscript, the authors should analyse their study design and their results.

Minor concerns

Material and Methods

This section needs more reference support.

1.- In section 2.2.- The authors should describe the feeding protocol for the rabbits and the biochemical parameters between groups of rabbits (heavy and light).

2.-  Transcriptome sequencing should be detailed in method section.

3.- Table 1 and Table 2 should be included in supplementary material.

4.- In section 2.7.- The authors did not define the antibodies.

5.- In section 2.10.- The quantification of Oil Red O staining was not detailed.

6.- An in silico section should be included and detailed by the authors. Prediction of target genes and selection to miR-383-5p.

Results

1.- In Section 3.3 .- The Figure 3E should be complemented with graph in box plot.

2.- In Section 3.4 .- The Figure 4E should be complemented with graph in box plot.                   

3.- In Section 3.5 .- The authors have previously showed miR-383-5p associated to lipid metabolism, but what is the rationale of treatment with PA as hypertrophic cells model?

4.- In Section 3.6 .- The authors study miR-383-5p regulation by RAD51AP1, but not CNIH4 and NUF2. The authors should rationale and detail this exclusion.

Comments on the Quality of English Language

Minor editing of English language required

Author Response

(The authors gave the same response as above.)

Reviewer 2 Report

Comments and Suggestions for Authors

The authors studied the role of miR-383-5p, a known regulator of cell proliferation and development, in adipogenesis. Its impact in such processes as preadipocytes proliferation and differentiation was assessed. An extensive transcriptome analyses was performed identifying potential genes associated with miR-383-5p, which proved involvement of miR-383-5p in regulation of lipogenesis, protein catabolism and glucose metabolism. During target gene prediction and dual luciferase reporter assay RAD51AP1 was identified as a target of miR-383-5p through which it mediates its effects.

The methodological level of the work is exceptionally high. The work is well illustrated with figures and tables.

There are minor improvements which may benefit the paper:

1.       Line 110 – NC and INC need deciphering

2.       Was test for normality of data performed during statistical analysis? Should be described.

3.       Lines 169-175 – red color of the script should be removed

4.       In methods the composition of high-fat diet should be described.

5.       Brief description of algorithms for target gene predictions may be represented as in  https://www.ncbi.nlm.nih.gov/pmc/articles/PMC5314757/

6.       Methods for detection of triglycerides and cholesterol in rabbits are not described in methods. How relative concentration was calculated? Using term “expression” for lipid fractions is not appropriate. Line174: The results of triglyceride (TG) and cholesterol (TC) measurements showed that heavy rabbits were significantly higher than light rabbits – sentence requires correction (…had higher level?? Of TG and TC???).  The same is applicable for lines 287-289.

7.       Figure 1D – it is not clear differences compared to which group are indicated by asterisks. Just one blue column is present in Fig 1D – is it combined data for both light and heavy animals?

8.       In discussion information on the physiological function of RAD51AP1 and scarce, but available information on its role in adipogenesis may benefit understanding the significance of the results. Possible sources for information:

https://www.sciencedirect.com/science/article/pii/S1097276522009352

https://www.ncbi.nlm.nih.gov/pmc/articles/PMC8248092/

https://www.ncbi.nlm.nih.gov/pmc/articles/PMC7854529/

Comments on the Quality of English Language

English is good, but there are minor mistakes in time tense, which occasionally are found through the text. 

Author Response

(The authors gave the same response as above.)
